# Public and patient perspectives on the use of clinical and administrative health data to identify and contact people at risk of future illness—The case of chronic kidney disease

**Donald J. Willison** [1] *, **Danielle M. Nash** [2,3,4], **Sarah E. Bota** [2,3], **Samar Almadhoun** [2,3], **Teresa Scassa** [5], **Amit X. Garg** [2,3,4,6], **Kidney Patient and Donor Alliance of Canada** [¶], **Ann Young** [2,7]

1 Institute of Health Policy, Management and Evaluation, University of Toronto, Toronto, Ontario, Canada, 2 ICES, Toronto, Ontario, Canada, 3 Lawson Health Research Institute and London Health Sciences Centre, London, Ontario, Canada, 4 Department of Epidemiology and Biostatistics, Schulich School of Medicine and Dentistry, Western University, London, Ontario, Canada, 5 Faculty of Law, Common Law Section, University of Ottawa, Ottawa, Ontario, Canada, 6 Division of Nephrology, Department of Medicine, Schulich School of Medicine and Dentistry, Western University, London, Ontario, Canada, 7 Division of Nephrology, Unity Health and the University of Toronto, Toronto, Ontario, Canada

¶ Representatives of the Kidney Patient and Donor Alliance of Canada are listed in the Acknowledgments.
* don.willison@utoronto.ca

**Data Availability Statement:** Within the manuscript, we have provided extensive quotes

## Abstract

For decades, researchers have used linkable administrative health data for evaluating the health care system, subject to local privacy legislation. In Ontario, Canada, the relevant privacy legislation permits some organizations (prescribed entities) to conduct this kind of research but is silent on their ability to identify and contact individuals in those datasets. Following consultation with the Office of the Information and Privacy Commissioner of Ontario, we developed a pilot study to identify and contact by mail a sample of people at high risk for kidney failure within the next 2 years, based on laboratory and administrative data from provincial datasets held by ICES, to ensure they receive needed kidney care. Before proceeding, we conducted six focus groups to understand the acceptability to the public and people living with chronic kidney disease of direct mail outreach to people at high risk of developing kidney failure. While virtually all participants indicated they would likely participate in the study, most felt strongly that the message should come directly from their primary care provider or whoever ordered the laboratory tests, rather than from an unknown organization. If this is not possible, they felt the health care provider should be made aware of the concern related to their kidney health. Most agreed that, if health authorities could identify people at high risk of a treatable life-threatening illness if caught early enough, there is a social responsibility to notify people. While privacy laws allow for free flow of health information among health care providers who provide direct clinical care, the proposed case-finding and outreach falls outside that model. Enabling this kind of information flow will require greater clarity in existing laws or revisions to these laws. This also requires adequate notification and culture change for health care providers and the public around information uses and flows.

from focus group participants. However, the transcripts themselves cannot be shared publicly, as this would violate the agreement to which the participants consented. In this revision, we have added a third appendix that provides a tabular summary of the themes that emerged from the focus groups and that served as the foundation for our analyses.

**Funding:** AG holds the Dr. Adam Linton Chair in Kidney Health Analytics (no grant number) https://www.schulich.uwo.ca/about/news/2019/october/announcement_dr_amit_garg_has_been_reappointed_as_the_endowed_research_chair_in_better_kidney_health.html AY is supported through a CIHR Health Research Training Award (MFE 171236) https://cihr-irsc.gc.ca/e/50513.html and a KRESCENT Post-Doctoral Fellowship (2020KP-PDF707719) https://kidney.ca/Krescent/Home The project was funded through a Canadian Nephrology Trials Network (CNTN) call for financial assistance. (No specific grant number) PIs Young and Garg The sponsors played no role in the study design, data collection and analysis, decision to publish, or preparation of the manuscript.

**Competing interests:** The authors have declared that no competing interests exist.

## 1. Introduction

For decades, researchers in numerous countries have had access to linkable administrative health data for analytics related to the health care system, subject to local privacy legislation and other data governance. To date, a common condition of access has been that the researcher does not attempt to contact individuals in the dataset. This condition is typically applied because of legislated health privacy consent requirements and strict limits on the use of health data for non-consented purposes. Any attempt to contact the patient would require express consent. Typically, this would be obtained by a representative of the organization that collected the information in the first place–for example, the health care provider.

ICES is a not-for-profit research institute in Ontario, Canada, whose researchers securely access the province's clinical and administrative health data to evaluate health care delivery and outcomes. ICES' authority to collect and link these data is through its special legal status as a "prescribed entity" under Ontario's Personal Health Information Protection Act (PHIPA) [1]. PHIPA and its regulations do not *explicitly* permit or prohibit a prescribed entity (such as ICES) to contact individuals. Accordingly, ICES has, thus far, not contacted individuals in the datasets being used for research and health care evaluation.

A team of investigators at ICES proposed a research initiative called KidneyCare Outreach to explore the possibility of using clinical and administrative health data held by ICES to identify and contact individuals at high risk of kidney failure who are not receiving guideline recommended care. The outreach initiative, consisting of a real-time assessment of their chronic kidney disease (CKD) and referral to a local nephrologist, is designed to serve as a health system safety net to help reduce their future risk for kidney failure.

When the investigators consulted with the Office of the Information and Privacy Commissioner, the Office expressed the view that sections of PHIPA could permit ICES to reach out to individuals identified in the dataset as being at risk for kidney failure, notify them of the research study, and seek their consent to participate in a research study (personal communication, February 18, 2021). However, they felt that direct communication with the health care provider who ordered tests that indicated an individual was at high risk of kidney failure would constitute disclosure of personal health information, requiring prior consent of the individual.

Based on the view expressed by the Office of the Information and Privacy Commissioner, the investigators developed a pilot research study to identify and contact a sample of people by mail who have evidence of being at high risk for kidney failure, based on individual-level demographics and laboratory test results from provincial datasets housed at ICES and health records indicating no prior contact with a nephrologist.

Before proceeding with the pilot study, we wanted to better understand the perspectives of the public and people living with CKD on (a) the acceptability of direct outreach by mail to people who may not be aware that they have CKD or are at high risk of kidney failure based on health administrative data, and (b) the conditions under which this outreach would be acceptable. This paper presents the results of focus groups that explored three main research themes:

- What factors would improve the likelihood of opening the mail invitation? What are participants' responses to specific aspects of the letter? What is the likelihood that they would participate in the outreach initiative?

- How acceptable is direct outreach to at-risk individuals for the purposes outlined in the study?

- Under what conditions would it be acceptable to expand this model of case-finding and outreach beyond research to routine population health management?

CKD is a good prototype to pilot a case finding and outreach program for several reasons: (a) It has a significant health impact, with a prevalence of 10% of the Canadian population and the tenth-leading cause of death in 2018 [2]; (b) one in four patients with kidney failure in 2020 in Canada were assessed by a nephrologist within only three months of first needing treatment for kidney failure [2], indicating that there is room for improvement; (c) Earlier detection of kidney disease is possible through routine laboratory tests that are readily available for analysis; and (d) There are evidence-based treatments available to prevent CKD from progressing to kidney failure, where the latter requires chronic dialysis or a kidney transplant to sustain life [3, 4].

## 2. Materials and methods

### 2.1 Deliberative focus group meetings

We conducted six virtual focus groups over seven weeks between January 21 and March 4 of 2023, using the "deliberative discussion focus group" method of Rothwell [5]. In a conventional focus group, little to no emphasis is placed on informing or educating the focus group member about the subject to be discussed. By contrast, the deliberative focus group method borrows from the deliberative discussion method of Fishkin and Luskin [6] by providing balanced and comprehensive information about the topic prior to deliberative discussion. In our focus groups, the educational components were introduced prior to each new theme raised. The deliberative focus groups were conducted virtually using Microsoft Teams because of the ongoing COVID-19 pandemic. All participants had to be 18 years or older, live in Ontario and be eligible for provincial health benefits. Focus group participants would be given an honorarium in the form of a $150 prepaid Amazon gift card.

Three of the focus groups involved members of the "general public". These participants were recruited by the Institute for Social Research (ISR) through a web-based survey using a consumer panel maintained by Leger [7, 8]. The ISR provided us with two lists of names. In the first wave, 13 of 22 respondents were 65 years or older and there were very few under the age of 35. We requested additional names. In the second wave, we received 40 names, all under the age of 65. Thus, the ISR provided us with a total of 62 names of people willing to participate. All were contacted, of whom 17 were available on the dates scheduled for the focus group meetings.

The other three focus groups consisted of individuals who had lived experience with CKD who were recruited by the Transplant Ambassador's Program using a web-based survey sent to their network contacts and posted on their social-media account [9]. Sixty-three people responded to the survey and social media posting. There was higher representation of individuals in the 18 to 24 years and 25 to 35 years age brackets than would be expected to be living with CKD. These respondents were traced back to the social media posting. We suspected these might be "imposter participants" seeking the $150 Amazon gift card honorarium being offered for focus group participation (n = 13 of the 63 respondents) [10]. Consequently, we limited our contacting to the 50 respondents who were known to the Transplant Ambassador's Program. Twenty-two of these were available on the scheduled dates. Participants recruited through Leger and the Transplant Ambassador's Program were directed to a website where the letter of information and informed consent were administered electronically through REDCap [11].

Focus groups ran for approximately three hours on a Saturday or Sunday. Meetings were co-moderated by two of the investigators (DJW and DMN). One moderator summarized points made by participants directly on a PowerPoint presentation during the meeting to allow for real-time member feedback, while the other took additional notes (not observed by

focus group participants), monitored any chats and coordinated the technical functions. Meetings were recorded and transcribed verbatim using Microsoft Teams. The number of participants in each focus group ranged from five to nine. The moderators concluded that saturation had been reached, as no substantive new themes emerged during the third focus groups for each of the general public and CKD meetings.

Meetings followed an interview guide involving presentation of material and structured but open-ended questions of participants. The interview guide was crafted by DJW and DMN to reflect operational responses to the three research themes identified above. Questions were reviewed and revised in consultation with the other research team members. The resulting questions were pilot tested with our patient partners who are also involved in the broader KidneyCare Outreach initiative. A summary of the focus group interview guide is outlined in **Box 1** and the full guide is available as **S1 Appendix**.

---

### Box 1. Summary of the deliberative focus group interview guide

- Following introductions and meeting goals, participants were shown a video of someone picking up their mail with the invitation letter amongst all the other letters and flyers in the mailbox, sorting through the mail, and pausing at the envelope with the study invitation.

- Participants then discussed the likelihood that they would open the letter and reasons why or why not. They were asked what would make the envelope more inviting to open.

- The video resumed with the individual opening the envelope and reading the letter.

- Participants were given the opportunity to read through the letter and FAQ document.

○ The personalized letter described kidney disease and its trajectory. It then indicated that the recipient's recent kidney test results suggested the need for further attention. It then described the KidneyCare Outreach project and invited the recipient to participate in the research.

- Participants were then asked their understanding of what the letter was asking them to do, what was going through their minds when reading it, and how they would respond to the invitation.

○ Mid-way through the focus groups, the invitation letter was revised based on the accrued feedback. Therefore, three groups reviewed the initial draft and the remaining three reviewed the revised draft letter.

- Up to this point, participants were deliberately kept naïve to the protocol details and the investigators. This was done to simulate the prior knowledge they would have if they were to receive the invitation themselves, unaware of what was behind it. In this next phase:

○ Participants were provided background information on ICES, its mandate, and how it uses and safeguards data. They were then given the opportunity to ask questions about ICES and provide their thoughts on ICES using health data collected during service delivery for managing the healthcare system and for health research.

---

○ They were then introduced to the idea of a "new" use of data held at ICES to identify individuals who may have a severe treatable illness and reaching out to them.

○ The case was made for CKD as a pilot project, with background information on severity of CKD and importance of early detection through laboratory testing. They were then informed how individuals who may have CKD would be identified and contacted by ICES.

• Participants were then asked:

○ What they liked about what the researchers planned to do;

○ What concerns they may have about this;

○ How they feel about an organization they do not know contacting them directly by letter to participate in the study; and

○ What would make this more acceptable?

• Participants were then given the opportunity to make additional comments about the envelope and letter in light of their more complete knowledge of the outreach initiative.

• We then probed the generalizability of people's views by asking how their responses would change if:

○ It was a less serious, but still treatable, condition;

○ There was no way to prevent the outcome through early detection;

○ This were to expand to a wide variety of health conditions?

• Finally, we re-framed the scenario and asked if the Province had a moral responsibility to do this type of case finding and outreach if it was able to identify people at risk of a life-threatening illness that could be treated or prevented if caught early enough.

The video used during the focus groups was initially drafted by DJW and DMN and revised in consultation with the rest of the research team and through our patient partner pilot. The envelope was drafted by SA and the initial draft of the letter by AY with guidance from the Office of the Information and Privacy Commissioner and in consultation with CKD care providers and patient partners. Amendments were made to the letter following the third focus group in response to the feedback received in the first three focus groups. This provided the opportunity to test the acceptability of the re-worded letter in the remaining three focus groups.

## 2.2 The case finding and contacting protocol

Under the protocol, a pre-defined algorithm was created to identify individuals at high risk of kidney failure based on existing laboratory data, with no evidence of recent contact with a nephrologist. Additional exclusions were applied to focus on individuals who could reasonably participate in a kidney outreach initiative. In this pilot phase, ICES will re-identify a small

subset of eligible individuals for potential contact to confirm the feasibility of the KidneyCare Outreach program's procedures. This personal health information will include names, addresses, telephone numbers, and relevant clinical and health services utilization data necessary for case identification. Personalized letters will be mailed to identified individuals by ICES staff with an invitation to participate in the pilot study. They may reply by telephone or via a secure website. If there is no response after two weeks, an individual from ICES will follow up by telephone.

The letter of invitation indicates: (a) how the individual was identified, (b) the nature of the organizations involved in the project (ICES and Kidney Care Outreach research team), (c) details about the outreach intervention, (d) how to find out more information and consent to participate, and (e) frequently asked questions. Brochures from the Kidney Foundation are also included in the envelope [12]. Digital copies of the letter and envelope are available as online **S2 Appendix**.

Those who agreed to participate complete a health questionnaire, have repeat lab tests, including serum creatinine and urine albumin-to-creatinine ratio, have a virtual visit with a study nephrologist, and are then referred to a local nephrologist for ongoing kidney care.

### 2.3 Ethics review

The KidneyCare Outreach pilot study and the focus group protocol on which we report here were approved by the Western University's Health Sciences Research Ethics Board (HSREB): File: 120319 (pilot study) and 121978 (focus groups).

### 2.4 Analysis

We conducted a thematic analysis of the responses to the focus group questions. The two moderators (DJW and DMN) summarized and analysed the research findings based on notes taken during the focus groups and the recordings. Notes from each focus group were entered into a spreadsheet organized by the questions asked with columns summarizing the general public and CKD focus groups separately, along with exemplar quotes extracted from the recordings. Summary notes were prepared in the four days following each focus group by DMN. The summary notes were discussed between DMN and DJW within seven days of each focus group. There was good concordance between DMN and DJW as to the messages coming from the meetings. Revisions to the summary notes were mainly regarding additional comments to include from the focus group discussion. There were no disagreements requiring a third assessor from among the team to adjudicate. After all focus groups were completed, a summary table was assembled. The table consolidated similar comments, allowing us to identify messages that occurred across all groups, most groups or some groups. All study team members reviewed the summary table and provided their feedback.

## 3. Results

### 3.1 Focus group participants

Between January 21st and March 4th, 39 people participated in six virtual deliberative focus groups (17 among the three general population focus groups (GP-FGs) and 22 among the three CKD-focus groups (CKD-FGs) (**Table 1**). The age distribution was balanced for the CKD-FGs but skewed toward older people in the GP-FGs. For the CKD-FGs, about half the participants indicated they had CKD but not kidney failure and the other half were receiving chronic dialysis or had a prior kidney transplant.

**Table 1. Characteristics of study participants.**

|  | General Public | Chronic Kidney Disease |
|---|---|---|
| **Total N Participants** | 17 | 22 |
| **Age Groups (years)** |  |  |
| 18–24 | 0 | 4 |
| 25–34 | 1 | 4 |
| 35–44 | 2 | 3 |
| 45–54 | 4 | 4 |
| 55–64 | 3 | 1 |
| 65+ | 7 | 6 |
| **Sex** |  |  |
| Female | 6 | 10 |
| Male | 11 | 12 |
| **Geographical Region** |  |  |
| Toronto | 3 | n/a |
| Other Greater Toronto Area | 4 | n/a |
| Eastern Ontario | 4 | n/a |
| Northeastern Ontario | 3 | n/a |
| Southwestern Ontario | 3 | n/a |
| **Patient Status** |  |  |
| Chronic kidney disease* | n/a | 10 |
| Chronic dialysis | n/a | 6 |
| Kidney transplant recipient | n/a | 6 |

*Not receiving treatment for end-stage kidney disease (including chronic dialysis or kidney transplant)

We intended to compare and contrast views of those living with CKD with views of the general public. Apart from minor differences in what would attract them to open the invitation envelope, we found virtually no difference in responses between the CKD focus groups and the general public focus groups.

## 3.2 Thematic analysis

A tabular summary of the themes that emerged from the focus groups may be found in S3 Appendix. (To protect identities, names have been redacted.) Below we provide a narrative description.

**3.2.1 The invitation itself (envelope and letter).** Almost all the participants across focus groups indicated that they would likely open the envelope. The most common reason cited for opening was that it looked official and legitimate. In particular: (a) the letter was addressed to them specifically; (b) the envelope itself was over-sized and stiff cardboard; and (c) it was not colourful, nor did it have anything that looked like advertising on the outside. In three focus groups, at least one participant remarked that the "Do not bend" instruction on the envelope piqued their interest.

Although participants did not generally recognize the name KidneyCare Outreach, the fact that the letter was coming from a kidney organization contributed to the interest in opening the letter in all three CKD focus groups (CKD-FGs). This was not raised in any of the general public focus groups (GP-FGs).

Several people in both groups remarked that there was no concern that it was spam or a scam.

"If I received this in the mail, I would open it because there's no danger of opening it. If I receive this as an e-mail, it would go to trash immediately."–Respondent 1, GP-FG1

The invitation letter itself was seen to be very professional-looking. They liked that the letters were signed by the principal investigator doctors with their hospital affiliations even though they might not be familiar with these hospitals.

In three focus groups (one CKD-FG and two GP-FGs), one or more participants indicated that the message in the text box immediately caught their attention. Indeed, several found the message very alarming:

"As soon as I saw that box–and I know this is only a focus group–but I didn't read the rest of the letter. I couldn't focus. I was just focused on that box. Oh no, my kidney. They found something with my kidney. I need more research. Oh, no. What's going to happen? What's the problem? And then I had a hard time reading the rest of the letter saying that they want to connect me with other services for further testing."–Respondent 2, GP-FG2

The wording in the original text box read: "Based on a recent lab test that you had done on ___, **you have been identified as a person who may be living with kidney disease.**" [bold text in the original]. Midway through the focus groups, this wording was revised to: "Based on a recent lab test that you had done on ___, **your kidney test results suggest the need for further attention**." (**S2 Appendix**) In the subsequent three focus groups, it was generally felt that the revised text struck a good balance:

"It wasn't alarming. It was very gentle, informative, it sort-of answered and anticipated all the questions I would have had if I were in that scenario." Respondent 3, GP-FG3

In four groups (two CKD-FGs and two GP-FGs), participants suggested that the letter be made clearer regarding how the organization obtained their personal health information. In two of the GP-FGs, participants indicated that they would want greater assurances about their privacy and the security of the data being used.

**3.2.2 Acceptability of direct outreach.** Participants expressed a strong preference that the researchers work directly with the provider who ordered the lab test and felt that the message should come directly from that provider, rather than from an organization they do not know:

"Why not go direct to the GP who ordered the lab result rather than direct to an individual, which suggests that you're bypassing the GP and potentially questioning the competency of the GP."–Respondent 1, CKD-FG3

"It comes down to a matter of trust. I've been with my family doctor for 16 years, and I trust that source of information versus some organization I have not heard from. So, they got to work together."–Respondent 2, CKD-FG3

In particular, because this is such a severe condition, it was suggested that the message should be given face-to-face or at least by telephone by a health care professional, so they could ask questions at the time.

"It's a really big deal to be told that you have kidney disease or potentially in some stage of kidney failure. It's not an easy thing. I think a lot of people see it as a sentence on your life."–Respondent 3, CKD-FG2

"It would cause me great concern if I knew nothing about kidney disease and all of a sudden, I see a physician's name signed off. I don't know who these doctors are and they're telling me that I might be at risk. So, it just feels isolated, out of context from my primary care."–Respondent 4, CKD-FG3

While virtually all participants indicated they would likely respond to the invitation to participate in the study, most participants would look more into the organizations involved–either reviewing their websites or calling them directly. Therefore, it was suggested that those who address public questions be made aware of this initiative, so they can prepare appropriate responses to any questions that may arise.

Many participants across both GP and CKD groups also stated that they would first contact the person who ordered the lab test to discuss the letter with them and they would then follow the advice of that provider.

"I might have some alarm to say I did not know I had that condition. That might be a reaction to it. And a second thought is I might want to consult my family doctor before taking action."–Respondent 4, GP-FG1

Some suggested copying the provider who ordered the lab test on the communication, so they would not be surprised when their patient contacted them:

"You've got to link the GP and the patient so that at least the GP knows that this may happen. The patient may get a letter, the patient may call the GP. There's got to be a whole communication mechanism around this initiative."–Respondent 1, CKD-FG3

Despite the strong exhortation to involve the family physician, the majority of participants saw the outreach program in a positive light. Some participants recognized that going through their healthcare provider may not be feasible given that healthcare providers are already overworked and that some things may get missed. They also recognized that some people do not have a primary care provider. Indeed, many of the participants who lacked a primary care provider or who lived remotely welcomed receiving such a letter:

"I would be thrilled that, oh, my gosh, somebody else is reaching out to me, especially up here in the Northwest, where medicine is much more difficult to get a hold of. I'd be thrilled to get an extra level of attention. And it would give me some hope and also the proactive feeling that was being projected into it."–Respondent 3, GP-FG3

"In the city where I live. . . we don't have a lot of doctors, so we rely on walk-in clinics. With KidneyCare Outreach, I would feel supported, I would be connected with counselors. I would be connected with support groups, everything I want to have with my family doctor and I don't. They would be offering it here in this specialized group in this specialized care directly for kidneys, and that would have me feeling supported, safe, heard, validated, that would give me what I need for that instance for what I'm going through."–Respondent 2, GP-FG2

"Sometimes waiting for a specialist can take an incredibly long time. And for us that live in smaller areas that may not have as many specialists residing in your area, this is a good way to start the ball rolling."–Respondent 5, GP-FG2

If direct mailing is the way individuals would be notified, participants in all groups suggested there be a general notice to the public indicating their information may be used to

identify and contact at-risk individuals. In four groups (2 CKD-FGs and 2 GP-FGs), several participants indicated they would prefer to be asked for their consent to sharing their personal health information before they received the letter. However, all felt the minimum should be notification that they may receive a letter from this organization so this does not come as a surprise. Participants suggested that this advanced notification could come from the person who originally ordered the test or from advertisement in the lab at the time of testing.

Even though most of the feedback from participants was positive, one person indicated that they would be angry to receive a letter like this:

> "I would be livid to be approached like this. Who are these people? How did they get to see my medical records or test results without my permission, without my knowledge, without my consent? And I would think they were a bunch of creepy snoops, and I wouldn't want anything to do with them."–Respondent 6, GP-FG2

**3.2.3 Thinking beyond CKD.**   As a form of sensitivity analysis, we probed the robustness of the responses to other health conditions of different severity and preventability. Respondents generally expressed the same interest in being notified should the health condition be less severe but treatable. A minority indicated that they would not want to be notified if there was no way to prevent or treat the health condition. However, the majority did still want to be notified. Some stated treatment could become available in the future, and they could keep an eye out. Some identified the opportunity to participate in research that might help others in the future. Across the GP-FGs, participants indicated that this knowledge may still help them take steps needed to arrange their lives.

**3.2.4 Expansion from research project to population-wide program(s).**   Most of the groups acknowledged that, were this kind of outreach to transition into a population-wide program, there would need to be an important culture shift in how healthcare is delivered. In one group, there was discussion of the importance of getting the buy-in of family physicians by emphasizing how this supports them in their work.

> "You can probably get [family doctors] to buy into it much easier if you say this is intended to be potentially a province-wide program which is intended to be supportive of you and the work that you do and not to take away from the MDs or the nurse practitioners, but a supporting mechanism for them. Because, let's face it, the healthcare system today in Ontario is not in good shape. And we don't have enough doctors and there are an awful lot of people that don't have doctors. So, this may be just one way of trying to provide that level of care that is currently not being provided at all."–Respondent 1, CKD-FG3

It was also suggested that letters come from a single organization that was known to the public and seen to be a trusted source–for example, the Ontario Ministry of Health. They also expressed that there should be broad notice of this program and the possibility of receiving a letter. Some examples given included at the time of renewing one's driver's license or health card.

Across all groups, at least some participants remarked that this would be a good use of taxpayers' money. In one group, it was suggested that this should improve health outcomes in the province.

**3.2.5 An ethical imperative to notify?.**   There was a strong affirmative response to our last question: "If the Province has the ability to identify specific people in its databases who are at high risk of a life-threatening illness, and that illness could be treated or prevented if caught

early enough, do you think there is an ethical imperative or social responsibility to notify people?" Some even indicated it would be unethical not to:

> "I think that there is an ethical imperative for sure and an obligation to let people know. If you don't know, you can't do anything. If you know and you don't do anything–well, OK, that's your choice. But at least you're given a choice." Respondent 6, CKD-FG2

Participants generally recognized that it all comes down to how such a program would be implemented. Who is the messenger? How does this fit into the health care system? In two focus groups (one each of GP-FG and CKD-FG), some participants identified that they were conflicted between privacy concerns and wanting to know this information.

## 4. Discussion

Case finding to identify and reach out to individuals who are at risk for future illness but not receiving recommended care is not unique. Data from various healthcare provider sources, such as hospitals, clinics, or independent health facilities have been used to identify undiagnosed chronic obstructive pulmonary disease in primary care [13, 14], undiagnosed type 2 diabetes from emergency department records [15], or communicable diseases like Hepatitis C virus [16]. In these cases, investigators reached out in collaboration with care providers known to those identified as being at risk. Clinical data may be further combined with data collected primarily for administrative and billing purposes (i.e., health administrative data) for a more comprehensive picture of patient health and processes of care [17].

Our study presents a novel approach to case finding, wherein province-wide health administrative data held by ICES, an organization that does not have direct connections with the data subjects, are being used in research to identify and reach out to individuals at high risk of kidney failure who are being missed. The ICES data repository contains data on over 14 million people living in Ontario, Canada [18], far exceeding the data available in any one or multiple networked electronic medical records. This would also be the first attempt at direct patient contact from reidentification of individuals within these province-wide health administrative databases.

In this study, we found that focus group participants were receptive to being contacted should they be potentially at-risk for a serious condition that could be reliably detected. Thoughtful implementation was felt to be paramount, including getting the "right" balance of conveying urgency while not inducing alarm when contacting at-risk individuals.

A common point of contention raised across focus groups was in the proposed approach to contacting at-risk individuals independent of primary care or other care providers known to them. In our pilot study, based on the views expressed in the letter from the Office of the Information and Privacy Commissioner of Ontario, investigators plan to reach out directly to at-risk individuals, and obtain their permission before reaching out to their primary care provider or ordering physician. The Office of the Privacy Commissioner was strong in its view that any contact between ICES and the primary care provider would not be permitted under Ontario's health information privacy law without prior consent by the individual. This is because contacting the primary health care provider with this information would constitute a disclosure of personal health information without consent, which the law prohibits. There are at least two possible scenarios in which this would be the case: First, the person who ordered the tests may not have been the primary health care provider. For example, it may have been a physician in a walk-in clinic or an emergency room. That physician may no longer be involved in the patient's care. Second, even if the notice went to the doctor who ordered the tests, these

tests may have been ordered for some other specific purpose, interpreted by the doctor and the patient treated. If the doctor did not seek out the broader CKD issues indicated by the tests, then there is a strong argument that it would be a disclosure if the researchers arrive at this interpretation and communicate it to the doctor.

While almost all the focus group participants indicated that they would respond positively if they were to receive the mail invitation to participate in the study, most expressed concern that ICES would not initially contact or work through their primary care provider or the ordering physician in this outreach initiative. They would prefer to be contacted directly by their primary care provider rather than by an organization with which they had no prior relationship. This is corroborated by evidence that participation in cancer screening improves when individuals are contacted through one's family physician [19]. Alternatively, it was suggested that the primary care provider be copied on the correspondence to the individual. To address the desire to keep the primary care provider in the loop while complying with the law, we now plan to include with the invitation letter a separate explanatory letter that the recipient can pass along to their primary care provider when they see them to discuss the letter.

Focus group participants in all groups suggested there be a general notice to the public indicating their information may be used to identify and contact at-risk individuals, so this does not come as a surprise. Several participants also indicated they would prefer to be asked for their consent to sharing their personal health information before they received any outreach. If obtaining consent was not feasible (as would be the case for a data repository containing information on over 14 million individuals), notification was still important. This raises several policy and process issues. (1) Currently, no mechanism exists to provide such notices. Should a notification system be developed, it would make sense to come from one authoritative source, such as the provincial health authority. This notice could be issued with the renewal notice for one's provincial health insurance card or driver's licence, for example. (2) Consideration needs to be given as to whether individuals would have the opportunity to opt-out of receiving such communications. (3) To minimize public confusion or concern over spam communication, outreach letters should also come from a single authoritative source.

The other major policy message we explored was that the proposed ICES initiative is research rather than a public or population health initiative of the government. From the perspective of our focus group participants, this issue seemed immaterial. Participants expressed that it is reasonable for public programs to start out as research initiatives and this distinction would not change their willingness to participate. It is not uncommon for population health services to begin as research studies. For example, having personalized screening reminders endorsed by an individual's family physician integrated into organized colon cancer screening in Ontario, Canada was first piloted in a research study for technical feasibility [20, 21].

Algorithms applied to population-wide clinical and administrative health data held by data institutes like ICES make it increasingly feasible to identify individuals with clinically important health concerns that may escape the attention of the clinician when reviewing an individual health record. In this study, we focused on the case of identifying individuals with advanced CKD at high risk for kidney failure. Algorithms could be similarly created to identify those with earlier-stage CKD, where earlier intervention may maximize benefit, or subsets of kidney disease patients, such as those with undiagnosed glomerulonephritis, where early identification may lead to treatment and cure. The same could also be applied to other disease entities with reliable biomarkers, for example, using HbA1c as an indicator for glycemic control in adults with diabetes who are not connected with endocrinologists or are not receiving guideline recommended therapies.

Case finding within large datasets may become more prominent as risk prediction models use even more sophisticated techniques, such as the application of deep learning algorithms

[22, 23]. Expansion would require careful scrutiny in the vetting and ongoing oversight of the algorithms being used to maximize population-level benefit, along with routine updates to address concept drift [24].

This also invites the question whether it should become public policy to routinely communicate relevant information to affected individuals and the conditions under which this should occur. Our focus group participants recognized that, if such public outreach programs were to be expanded, it may be impracticable for them to be run through individuals' primary care providers. In that case, they indicated primary care providers should be kept in the loop in both developing these programs and in the implementation of any program. Further, if such outreach programs were not initiated by the primary care providers, they recommended: (1) that all such initiatives should come from one trusted source with clear communication to the public that these programs function with authority in law; and (2) that this communication be part of a comprehensive public communication about the broader uses of personal health information beyond direct clinical care.

Achieving this transition to the use of administrative health data for population health management will require reform to existing privacy laws, as most were developed prior to the capacity to algorithmically identify at-risk individuals as part of a population health initiative. While privacy laws generally allow for free flow of health information among health care providers who provide direct clinical care, this form of population health management falls outside that model of health care. We suggest that legislators consider amending their privacy laws to permit entities like ICES, under controlled circumstances, to conduct these types of research and population health initiatives and to communicate their findings to primary care providers, as if they were part of the "circle of care" and possibly also to at-risk individuals directly. This would also require initiatives to address the culture change around acceptable information use and flows for health care providers and the public, as recommended by our focus group participants.

Further, it is difficult for privacy laws to keep pace with the changes in information flows associated with algorithmic use of personal health information for population health purposes. Possible interim solutions include: (a) promulgation of clear published guidance from information and privacy commissioners to assist researchers with designing compliant research studies; and (b) development of a regulatory sandbox–i.e., a safe space where organizations can test uses of personal data in innovative ways, in a controlled environment. This has been done in other jurisdictions, though not yet in the context of personal health information [25, 26].

### 4.1 Limitations

As part of our inclusion criteria, we did not ask about level of education. However, as we found during our meeting introductions, participants in both the CKD and the general public groups in this study were well-educated, with many having professional careers. Further, they demonstrated a high level of health-consciousness. While limiting our insights into the perspectives of those with less formal education and less health-conscious, this permitted a very rich discussion.

### 5. Conclusion

Innovations in the capacity to manage and analyse personal health information at a population level allow us to move beyond using these data for health systems management and research to improving the care of individuals at risk of future illness, either through their primary care provider or direct outreach to at-risk individuals. Our focus group work suggests that the

public are very open to this possibility so long as primary care providers are kept in the loop, it is done through trustworthy institutions (e.g., government health authorities), and there is proper notification to the public as to the legitimacy of these uses. Such a program would potentially circumvent or delay the onset of debilitating illness and, therefore, improve population health.

Any program of this nature will require greater clarity in existing laws or revisions to these laws to address the circumstances under which these uses are permitted. Also, there is a need for mechanisms within legislation (e.g., regulatory sandboxes) to permit innovative uses of health information in a timely fashion. All this will contribute to the capacity to run effective learning health systems.

## Supporting information

**S1 Appendix. Full interview guide for focus groups.**
(PDF)

**S2 Appendix. Draft invitation letter presented to focus groups.** Note this is the second version of the letter presented to the last three focus groups.
(DOCX)

**S3 Appendix. Thematic summary of focus group findings.**
(DOCX)

## Acknowledgments

We would like to acknowledge our patient partners (all affiliated with the Kidney Patient and Donor Alliance of Canada), including:

Susan McKenzie (lead, susan@kidneyalliance.ca),
Patricia Kay,
Peter Wechselmann,
James Baird,
Dale Bouskill

Parts of this material are based on data and information compiled and provided by the Institute for Social Research. The analyses, conclusions, opinions and statements expressed herein are solely those of the authors and do not reflect those of the funding or data sources; no endorsement is intended or should be inferred. We would also like to thank Craig Lindsay for his contributions to the KidneyCare Outreach initiative.

## Author Contributions

**Conceptualization:** Donald J. Willison, Teresa Scassa, Ann Young.

**Data curation:** Danielle M. Nash.

**Formal analysis:** Danielle M. Nash.

**Funding acquisition:** Amit X. Garg, Ann Young.

**Investigation:** Donald J. Willison, Danielle M. Nash.

**Methodology:** Donald J. Willison, Teresa Scassa.

**Project administration:** Sarah E. Bota, Samar Almadhoun.

**Supervision:** Donald J. Willison, Amit X. Garg, Ann Young.

**Writing – original draft:** Donald J. Willison.

**Writing – review & editing:** Donald J. Willison, Danielle M. Nash, Sarah E. Bota, Samar Almadhoun, Teresa Scassa, Amit X. Garg, Ann Young.

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
