## [Decision Letter · Decision Letter 0]

6 Nov 2023

PONE-D-23-30054Public and Patient Perspectives on the Use of Clinical and Administrative Health Data to Identify and Contact People at Risk of Future Illness – the Case of Chronic Kidney Disease.PLOS ONE

Dear Dr. Willison,

Thank you for submitting your manuscript to PLOS ONE. After careful consideration, we feel that it has merit but does not fully meet PLOS ONE’s publication criteria as it currently stands. Therefore, we invite you to submit a revised version of the manuscript that addresses the points raised during the review process.

The research question addressed by authors is very interesting. The novelty, sound methodology and important public health implications make this manuscript appropriate for publication in PLOS ONE. However, before acceptance, we ask the authors to provide a revision of their manuscript with a point-by-point response to the two reviewers’ comments.In addition, please remove numbering from within the discussion and rather use firstly, secondly, etc…and please  add more references to support your discussion, as 4 references are insufficient and the discussion seems to interpret the findings without any comparison to the available literature.  Please submit your revised manuscript by Dec 21 2023 11:59PM. If you will need more time than this to complete your revisions, please reply to this message or contact the journal office at plosone@plos.org. Please include the following items when submitting your revised manuscript:A rebuttal letter that responds to each point raised by the academic editor and reviewer(s). You should upload this letter as a separate file labeled 'Response to Reviewers'.A marked-up copy of your manuscript that highlights changes made to the original version. You should upload this as a separate file labeled 'Revised Manuscript with Track Changes'.An unmarked version of your revised paper without tracked changes. You should upload this as a separate file labeled 'Manuscript'.

We look forward to receiving your revised manuscript.

Kind regards,

Mabel Aoun, MD, MPH

Academic Editor

PLOS ONE

Journal Requirements:

This study received funding from the Canadian Nephrology Trials Network and the New Frontiers in Research Fund. Parts of this material are based on data and information compiled and provided by the Institute for Social Research. The analyses, conclusions, opinions and statements expressed herein are solely those of the authors and do not reflect those of the funding or data sources; no endorsement is intended or should be inferred. We would also like to thank Craig Lindsay for his contributions to the KidneyCare Outreach initiative.

Dr. Amit Garg was supported by the Dr. Adam Linton Chair in Kidney Health Analytics. 

Dr. Ann Young was supported by post-doctoral fellowships from CIHR and KRESCENT.

AG holds the Dr. Adam Linton Chair in Kidney Health Analytics (no grant number) https://www.schulich.uwo.ca/about/news/2019/october/announcement_dr_amit_garg_has_been_reappointed_as_the_endowed_research_chair_in_better_kidney_health.html 

AY is supported through a CIHR Health Research Training Award (MFE 171236) https://cihr-irsc.gc.ca/e/50513.html and a KRESCENT Post-Doctoral Fellowship (2020KP-PDF707719)  https://kidney.ca/Krescent/Home

The project was funded through a Canadian Nephrology Trials Network (CNTN) call for financial assistance. (No specific grant number) PIs Young and Garg

The sponsors played no role in the study design, data collection and analysis, decision to publish, or preparation of the manuscript

3. One of the noted authors is a group or consortium Susan McKenzie, Patricia Kay, Peter Wechselmann, James Baird, Dale Bouskill. In addition to naming the author group, please list the individual authors and affiliations within this group in the acknowledgments section of your manuscript. Please also indicate clearly a lead author for this group along with a contact email address.

Reviewers' comments:

Reviewer's Responses to Questions

**Comments to the Author**

1. Is the manuscript technically sound, and do the data support the conclusions?

Reviewer #1: Yes

Reviewer #2: Partly

2. Has the statistical analysis been performed appropriately and rigorously? 

Reviewer #1: Yes

Reviewer #2: N/A

3. Have the authors made all data underlying the findings in their manuscript fully available?

Reviewer #1: No

Reviewer #2: No

4. Is the manuscript presented in an intelligible fashion and written in standard English?

Reviewer #1: Yes

Reviewer #2: Yes

5. Review Comments to the Author

Reviewer #1: The research project addresses a very important health topic on the possibility to use of administrative health data to contact people who might be at risk of developing serious conditions such as CDK, which could have major practical implications in terms of health policy and public health.

The authors conducted a qualitative study aiming to better understand the perspectives of the public and people living with CKD on (a) the acceptability of direct outreach by mail to people who may not be aware that they have CKD or are at high risk of kidney failure based on health administrative data, and (b) the conditions under which this outreach would be acceptable. Focus groups were conducted with members of the general public and people are risk of having kidney disease.

The research seems to be well conducted. Overall, the manuscript is clear and well-written. However, more information is needed for several sections of the manuscript:

- In the methods section, on page 3, the authors mention the use of “deliberative discussion focus groups”. This method should be explained further, and in particular the difference with traditional focus groups.

- 6 focus groups were conducted with 5 to 9 participants in each focus group. On page 4, the authors mentioned that members of the general public were recruited via a web-based survey using a consumer panel, and that the CDK group was recruited by the Transplant’s ambassador program. We need more information on the recruitment process for participants. How many people were contacted? How many responded? How were the final participants selected among those who were interested in participating?

- The interview guide is well structured, comprehensive, and relevant for the research objectives. However, there is no information in the manuscript on how this guide was developed. Who was involved in drafting the guide (Was it a cross-disciplinary team? Or was it only the two people leading the focus groups?)? How were the questions chosen? Similarly, what was the process to create the video content? What steps were taken to choose the visual for the envelop and the content of the information letter that are presented in the video?

- On page 7, the authors mentioned the use of an algorithm that was created to identify individuals with a ≥25% chance of kidney failure in the coming 2 years. They did not provide any information on this algorithm. More information should be provided to explain 1/ how the algorithm was developed and by whom, 2/ the specific criteria (clinical criteria?) used in the algorithm. This could be included in an appendix file.

- In the analysis section, on page 8, the authors describe qualitative analyses performed on the focus groups data. However, the exact method used for qualitative analyses is not mentioned (thematic analysis, grounded theroy...). Based on the information provided, it seems to be a thematic analysis. The authors should provide a clear statement explaining this point. In addition, the process used for data analysis is quite vague and could be explained more clearly. Authoors should aslo detail how disagreements among researchers in the analysis of focus group data were resolved.

- The discussion section is clear and very informative. It would be great to have a more in-depth discussion of the possibility to expand the project to other diseases (serious or less serious conditions, treatable and non-treatable conditions).

Reviewer #2: This article explores the acceptability of direct mail outreach to people at high risk of developing kidney failure, from the perspective of the public and of persons living with chronic kidney disease. It uses an approach involving focus group discussions and qualitative content analysis. I thank the authors for developing and elaborating a good manuscript on this interesting topic. I have some comments and suggestions that may be useful to improve the manuscript, for the authors’ consideration.

1. In the abstract, I suggest including the number of focus group discussions.

2. In the abstract, it is preferable to avoid using abbreviations (ICES). Also, in the introduction, the abbreviation ICES should be expanded.

3. Line 46, I suggest rephrasing the research question, avoiding the phrase ‘maximizing the likelihood’, and ’what is the likelihood’, as these sugest a quantitative investigation. For example, “what factors may affect the likelihood…”, or other.

4. Line 55, “One in four patients”.

5. In the limitations section of the discussion, I suggest avoiding a reference to generalizability, as that is not an aim of qualitative investigations. Instead, it would be relevant to describe the trustworthiness of the research, focusing on data credibility (e.g. used both general population and chronic kidney disease focus groups, range of ages among participants, and experience of investigators on the topic), dependability (e.g. the use of the interview guide) and transferability (e.g. described the context).

6. Line 429, please spell out what ‘this’ is.

7. Throughout the article, the word ‘will’ is regularly used (e.g. line 432). This word suggests certainty that this will happen. It may be more appropriate to replace this word with ‘would’ in many instances in the manuscript, to reflect a lower degree of certainty.

8. If available, I think it would be useful to have a table or figure providing an overview of the themes identified (referred to in line 89).

6. PLOS authors have the option to publish the peer review history of their article (what does this mean?). If published, this will include your full peer review and any attached files.

Reviewer #1: No

Reviewer #2: No

---

## [Author Response · Author response to Decision Letter 0]

5 Jan 2024

Editor comments:

Thank you for submitting your manuscript to PLOS ONE. After careful consideration, we feel that it has merit but does not fully meet PLOS ONE’s publication criteria as it currently stands. Therefore, we invite you to submit a revised version of the manuscript that addresses the points raised during the review process.

The research question addressed by authors is very interesting. 

The novelty, sound methodology and important public health implications make this manuscript appropriate for publication in PLOS ONE. However, before acceptance, we ask the authors to provide a revision of their manuscript with a point-by-point response to the two reviewers’ comments.

In addition, please remove numbering from within the discussion and rather use firstly, secondly, etc

Response: Done

…and please add more references to support your discussion, as 4 references are insufficient and the discussion seems to interpret the findings without any comparison to the available literature. 

Response: We have included additional content and supportive references in the Discussion section. See lines 362 to 383. (Note: line numbers here and throughout refer to the revised manuscript with tracked changes.)

We look forward to receiving your revised manuscript.

Kind regards,

Mabel Aoun, MD, MPH

Academic Editor

Journal Requirements:

https://journals.plos.org/plosone/s/file?id=wjVg/PLOSOne_formatting_sample_main_body.pdfand

Response: Done. As these were style requirements, we did not do these in track changes so as not to distract the reviewers. 

This study received funding from the Canadian Nephrology Trials Network and the New Frontiers in Research Fund. Parts of this material are based on data and information compiled and provided by the Institute for Social Research. The analyses, conclusions, opinions and statements expressed herein are solely those of the authors and do not reflect those of the funding or data sources; no endorsement is intended or should be inferred. We would also like to thank Craig Lindsay for his contributions to the KidneyCare Outreach initiative.

Dr. Amit Garg was supported by the Dr. Adam Linton Chair in Kidney Health Analytics. 

Dr. Ann Young was supported by post-doctoral fellowships from CIHR and KRESCENT.

Response: We have removed the funding information from the Acknowledgments. 

AG holds the Dr. Adam Linton Chair in Kidney Health Analytics (no grant number) https://www.schulich.uwo.ca/about/news/2019/october/announcement_dr_amit_garg_has_been_reappointed_as_the_endowed_research_chair_in_better_kidney_health.html

AY is supported through a CIHR Health Research Training Award (MFE 171236) https://cihr-irsc.gc.ca/e/50513.html and a KRESCENT Post-Doctoral Fellowship (2020KP-PDF707719) https://kidney.ca/Krescent/Home

The project was funded through a Canadian Nephrology Trials Network (CNTN) call for financial assistance. (No specific grant number) PIs Young and Garg

The sponsors played no role in the study design, data collection and analysis, decision to publish, or preparation of the manuscript

Response: No need to change the Funding Statement previously submitted. 

3. One of the noted authors is a group or consortium Susan McKenzie, Patricia Kay, Peter Wechselmann, James Baird, Dale Bouskill. In addition to naming the author group, please list the individual authors and affiliations within this group in the acknowledgments section of your manuscript. Please also indicate clearly a lead author for this group along with a contact email address.

Response: The authors within the group consists of our patient partners. Their affiliation is “Kidney Patient and Donor Alliance Canada” (they do not have academic faculty appointments). We have written in the acknowledgements as follows:

“We would like to acknowledge our patient partners (all affiliated with the Kidney Patient and Donor Alliance of Canada), including Susan McKenzie (lead, susan@kidneyalliance.ca), Patricia Kay, Peter Wechselmann, James Baird, Dale Bouskill.”

Response to reviewers:

Reviewer 1

The research project addresses a very important health topic on the possibility to use of administrative health data to contact people who might be at risk of developing serious conditions such as CDK, which could have major practical implications in terms of health policy and public health.

The authors conducted a qualitative study aiming to better understand the perspectives of the public and people living with CKD on (a) the acceptability of direct outreach by mail to people who may not be aware that they have CKD or are at high risk of kidney failure based on health administrative data, and (b) the conditions under which this outreach would be acceptable. Focus groups were conducted with members of the general public and people are risk of having kidney disease.

The research seems to be well conducted. Overall, the manuscript is clear and well-written. However, more information is needed for several sections of the manuscript:

Questions: 

1. In the methods section, on page 3, the authors mention the use of “deliberative discussion focus groups”. This method should be explained further, and in particular the difference with traditional focus groups.

Response: This is now explained on lines 68 to 73 in the revised manuscript with changes tracked. 

2. 6 focus groups were conducted with 5 to 9 participants in each focus group. On page 4, the authors mentioned that members of the general public were recruited via a web-based survey using a consumer panel, and that the CDK group was recruited by the Transplant’s ambassador program. We need more information on the recruitment process for participants. How many people were contacted? How many responded? How were the final participants selected among those who were interested in participating?

Response: We have added detail around the recruitment process for both groups to address the questions above. Please see lines 75 through 106 for revisions. 

3. The interview guide is well structured, comprehensive, and relevant for the research objectives. However, there is no information in the manuscript on how this guide was developed. Who was involved in drafting the guide (Was it a cross-disciplinary team? Or was it only the two people leading the focus groups?)? How were the questions chosen? Similarly, what was the process to create the video content? What steps were taken to choose the visual for the envelop and the content of the information letter that are presented in the video?

Response: We have clarified the process for developing the focus group questions on lines 118 to 121 of the revised manuscript. The process for developing the video is now described in lines 125 to 127 and the letter in lines 127-132. 

4. On page 7, the authors mentioned the use of an algorithm that was created to identify individuals with a ≥25% chance of kidney failure in the coming 2 years. They did not provide any information on this algorithm. More information should be provided to explain 1/ how the algorithm was developed and by whom, 2/ the specific criteria (clinical criteria?) used in the algorithm. This could be included in an appendix file.

Response: The algorithm was developed by the broader KidneyCare Outreach study team using data contained within provincial health administrative databases (e.g., laboratory data, demographics, and health services utilization). It will be used to identify which patients are at high risk, such that they can be contacted (the intervention). This manuscript focuses on public and patient perspectives to the use of health administrative data for kidney care outreach. Details on the algorithm and its development are out of the scope of this manuscript and will be published separately (currently under review). 

5. In the analysis section, on page 8, the authors describe qualitative analyses performed on the focus groups data. However, the exact method used for qualitative analyses is not mentioned (thematic analysis, grounded theory...). Based on the information provided, it seems to be a thematic analysis. The authors should provide a clear statement explaining this point. In addition, the process used for data analysis is quite vague and could be explained more clearly. Authors should also detail how disagreements among researchers in the analysis of focus group data were resolved.

Response: We have expanded the discussion of our analysis methods in lines 163 to 179 to address this. 

6. The discussion section is clear and very informative. It would be great to have a more in-depth discussion of the possibility to expand the project to other diseases (serious or less serious conditions, treatable and non-treatable conditions).

Response: We have included additional discussion and examples of how this type of project could be expanded. See lines 442 to 456.

Reviewer 2

This article explores the acceptability of direct mail outreach to people at high risk of developing kidney failure, from the perspective of the public and of persons living with chronic kidney disease. It uses an approach involving focus group discussions and qualitative content analysis. I thank the authors for developing and elaborating a good manuscript on this interesting topic. I have some comments and suggestions that may be useful to improve the manuscript, for the authors’ consideration.

1. In the abstract, I suggest including the number of focus group discussions.

Response: Done! 

2. In the abstract, it is preferable to avoid using abbreviations (ICES). Also, in the introduction, the abbreviation ICES should be expanded.

Response: In 2018, the institute formerly known as the ‘Institute for Clinical Evaluative Sciences’ formally adopted the initialism ICES as its official name. Thus, ICES is not an abbreviation, but the formal name of the institution in Ontario that evaluates healthcare delivery and outcomes. 

We agree, though, that it is best that we do not use the term “ICES” in the abstract. Because we were already at the limit of our word count, we chose to simply replace “ICES investigators” with “we”. Then, we explain what ICES is in the Introduction. 

3. Line 46, I suggest rephrasing the research question, avoiding the phrase ‘maximizing the likelihood’, and ’what is the likelihood’, as these suggest a quantitative investigation. For example, “what factors may affect the likelihood…”, or other.

Response: We have revised to “What factors would improve the likelihood…” as this is the way in which our questions in the focus groups were framed. 

4. Line 55, “One in four patients”.

Response: Done!

5. In the limitations section of the discussion, I suggest avoiding a reference to generalizability, as that is not an aim of qualitative investigations. Instead, it would be relevant to describe the trustworthiness of the research, focusing on data credibility (e.g. used both general population and chronic kidney disease focus groups, range of ages among participants, and experience of investigators on the topic), dependability (e.g. the use of the interview guide) and transferability (e.g. described the context).

Response: We removed our use of the word “generalizability”. Our change can be found on line 494 to 495.

6. Line 429, please spell out what ‘this’ is.

Response: We have made revisions to two mentions of “this” to remove the ambiguity in the Conclusions. See line 505 and line 508. 

7. Throughout the article, the word ‘will’ is regularly used (e.g. line 432). This word suggests certainty that this will happen. It may be more appropriate to replace this word with ‘would’ in many instances in the manuscript, to reflect a lower degree of certainty.

Response: Throughout the manuscript, in situations where there is a high degree of certainty, we have kept the “will” terminology. However, in other situations with a lower degree of certainty, we have revised to “would” accordingly. 

8. If available, I think it would be useful to have a table or figure providing an overview of the themes identified (referred to in line 89). 

Response: We have now added a third appendix which consists of the summary table that we used to inform our manuscript. We have also revised lines 196 to 198 to call the reader’s attention to the table.

---

## [Decision Letter · Decision Letter 1]

24 Jan 2024

Public and Patient Perspectives on the Use of Clinical and Administrative Health Data to Identify and Contact People at Risk of Future Illness – the Case of Chronic Kidney Disease.

PONE-D-23-30054R1

Dear Dr. Willison,

We’re pleased to inform you that your manuscript has been judged scientifically suitable for publication and will be formally accepted for publication once it meets all outstanding technical requirements.

Kind regards,

Mabel Aoun, MD, MPH

Academic Editor

PLOS ONE

Additional Editor Comments (optional):

Reviewers' comments:

Reviewer's Responses to Questions

**Comments to the Author**

1. If the authors have adequately addressed your comments raised in a previous round of review and you feel that this manuscript is now acceptable for publication, you may indicate that here to bypass the “Comments to the Author” section, enter your conflict of interest statement in the “Confidential to Editor” section, and submit your "Accept" recommendation.

Reviewer #1: All comments have been addressed

Reviewer #2: All comments have been addressed

2. Is the manuscript technically sound, and do the data support the conclusions?

Reviewer #1: Yes

Reviewer #2: Yes

3. Has the statistical analysis been performed appropriately and rigorously? 

Reviewer #1: Yes

Reviewer #2: N/A

4. Have the authors made all data underlying the findings in their manuscript fully available?

Reviewer #1: No

Reviewer #2: No

5. Is the manuscript presented in an intelligible fashion and written in standard English?

Reviewer #1: Yes

Reviewer #2: Yes

6. Review Comments to the Author

Reviewer #1: (No Response)

Reviewer #2: (No Response)

7. PLOS authors have the option to publish the peer review history of their article (what does this mean?). If published, this will include your full peer review and any attached files.

Reviewer #1: No

Reviewer #2: No

---

## [Editor Report · Acceptance letter]

22 Feb 2024

PONE-D-23-30054R1 

PLOS ONE

Dear Dr. Willison, 

I'm pleased to inform you that your manuscript has been deemed suitable for publication in PLOS ONE. Congratulations! Your manuscript is now being handed over to our production team.

Kind regards, 

on behalf of

Dr. Mabel Aoun 

Academic Editor

PLOS ONE